# Genetic Analyses of Response of Local Ghanaian Tanzanian Chicken Ecotypes to a Natural Challenge with Velogenic Newcastle Disease Virus

**DOI:** 10.3390/ani12202755

**Published:** 2022-10-13

**Authors:** Muhammed Walugembe, Augustine Naazie, James R. Mushi, George A. Akwoviah, Esther Mollel, Juliana A. Mang’enya, Ying Wang, Nadira Chouicha, Terra Kelly, Peter L. M. Msoffe, Hope R. Otsyina, Rodrigo A. Gallardo, Susan J. Lamont, Amandus P. Muhairwa, Boniface B. Kayang, Huaijun Zhou, Jack C. M. Dekkers

**Affiliations:** 1Department of Animal Science, Iowa State University, 2255 Kildee Hall, Ames, IA 50011, USA; 2Genomics to Improve Poultry Innovation Lab, University of California, Davis, CA 95616, USA; 3Department of Animal Science, University of Ghana, P.O. Box LG 25 Legon, Accra, Ghana; 4Department of Veterinary Medicine and Public Health, Sokoine University, P.O. Box 3000 Chuo Kikuu, Morogoro, Tanzania; 5Department of Animal Science, University of California, Davis, CA 95616, USA; 6School of Veterinary Medicine, University of California, Davis, CA 95616, USA; 7School of Veterinary Medicine, University of Ghana, P.O. Box LG 25 Legon, Accra, Ghana

**Keywords:** Newcastle disease virus, genetic parameters, disease exposure, local chicken ecotypes

## Abstract

**Simple Summary:**

Newcastle disease (ND) is a global threat to poultry production and often has a major impact on chicken welfare and the livelihoods of rural poultry farmers. We exposed unvaccinated Ghanaian and Tanzanian local chicken ecotypes to velogenic Newcastle disease virus strains, and measured response traits to understand the genetic basis of ND. We identified heritable ND response traits and revealed differences in survival between Ghanaian and Tanzanian local chicken ecotype birds. Our findings indicate that velogenic ND resistance could be improved through selective breeding of local chicken ecotypes in regions where the disease is endemic.

**Abstract:**

Newcastle disease is a devastating poultry disease that often causes significant economic losses in poultry in the developing countries of Africa, Asia, as well as South and Central America. Velogenic Newcastle disease virus (NDV) outbreaks are associated with high mortalities, which can threaten household livelihoods, especially in the rural areas, and lead to loss of high-quality proteins in the form of meat and eggs, as well as household purchasing power. In this study, we exposed unvaccinated Ghanaian and Tanzanian chickens of six local ecotypes to velogenic NDV strains, measured NDV response traits, sequenced their DNA on a genotyping-by-sequencing platform, and performed variance component analyses. The collected phenotypes included: growth rates (pre- and post-exposure); lesion scores (gross lesion severity) in the trachea, proventriculus, intestine, and cecal tonsils; natural antibody levels; anti-NDV antibody levels at 7 days post exposure (dpe); tear and cloacal viral load at 2, 4, and 6 dpe; and survival time. Heritability estimates were low to moderate, ranging from 0.11 for average lesion scores to 0.36 for pre-exposure growth rate. Heritability estimates for survival time were 0.23 and 0.27 for the Tanzanian and Ghanaian ecotypes, respectively. Similar heritability estimates were observed when data were analyzed either separately or combined for the two countries. Survival time was genetically negatively correlated with lesion scores and with viral load. Results suggested that response to mesogenic or velogenic NDV of these local chicken ecotypes could be improved by selective breeding. Chickens that are more resilient to velogenic NDV can improve household livelihoods in developing countries.

## 1. Introduction

In Africa, chickens are vital livestock to rural people’s livelihoods. They are a source of revenue and food through the provision of eggs and meat [1]. Most rural households raise chickens from local ecotypes, using free-range or backyard traditional extensive management systems [2,3]. Chickens also play an important role in gender empowerment because chickens are typically managed by women and provide income-generating avenues for women. Chicken farming in rural areas is characterized by low input and farmers often experience high mortality rates in their flocks due to poor husbandry practices and disease [4].

Newcastle disease virus (NDV), avian paramyxovirus type 1, belongs to the paramyxovidae family [5], and is one of the most important pathogenic viruses in poultry, especially in the developing countries of Africa and Asia. Due to viral evolution, there is high genetic diversity of NDV in both west and sub-Saharan Africa, with various genotypes of sublineages reported to be indigenous to Africa [6,7]. Based on their pathological type, strains of NDV are classified into lentogenic, mesogenic, and velogenic strains [8,9,10]. Velogenic strains cause severe disease and are endemic and responsible for high mortality in poultry in the developing countries of Africa [11,12,13], Asia [14], and South America. Outbreaks are often devastating, and a velogenic strain can cause up to 100% mortality in chicken flocks, resulting in severe economic losses for families. The use of vaccination to control NDV in backyard or scavenging chickens is challenging because of poor infrastructure to transport vaccines, the inability to maintain the required cold chain, and the high costs of the vaccine [15]. Although, vaccination may protect against clinical signs, it may not protect against viral shedding, depletion of lymphocytes, and atrophy [15]. The threat of highly virulent NDV continues in developing countries because the disease is endemic and there is limited financial and technical support from government agencies to local village poultry farmers [16]. As a complimentary approach to vaccination, genetic selection of local chicken ecotypes for resistance or resilience to NDV is a good strategy to limit the impacts of NDV on smallholders in developing countries.

Our previous studies on the response of local chicken ecotypes from Tanzania [17] and Ghana [18] to experimental infection with lentogenic NDV reported moderate heritabilities for various traits, including pre- and post-infection growth rate, anti-NDV antibody levels, and NDV viral load at 2- and 6-days post infection. However, there is limited literature on the genetic basis of the response of local African chickens to velogenic NDV infection. Experimental infection trials with velogenic strains are difficult and expensive because of the high biocontainment required. As a result, such experiments are often conducted with small sample sizes [19,20]. Routine large-scale experimental infection with Marek’s disease in a commercial layer breeding program have been conducted and have reported significant progress in reducing mortality from Marek’s disease [21]. In pigs, natural exposure to viral diseases has been utilized as a source of data for genetic analyses [22,23]. Natural disease exposure experiments usually require the use of seeder animals, allowing the determination of heritability for disease response traits [24,25], despite the potential uncontrolled nature of infections among individuals. Protocols for natural exposure of chickens to pathogenic NDV strains have been developed [26,27] and used on the local chicken ecotypes studied here [18,19]. These natural exposure studies, however, resulted in low mortality, likely because of the protective effects of prior inoculation with the vaccine strain of NDV, despite low remaining antibody levels at the time of the infection, and very low estimates of heritability of NDV response traits [26]. In the studies reported here, natural exposure trials were conducted with chickens from the same local ecotypes from Tanzania and Ghana as evaluated in [17,18,28], but without prior infection or vaccination with the lentogenic vaccine strain. Natural exposure trials were conducted rather than controlled experimental infection trials to mimic NDV disease environments on rural farms or in backyards, where NDV is endemic. The use of unvaccinated birds was designed to maximize expression of genetic differences in response to velogenic NDV. 

Against this background, the objectives of this study were to estimate genetic parameters (heritabilities and phenotypic and genetic correlations) for NDV response traits following natural exposure to birds that were naturally infected with virulent NDV strains, and to estimate genetic correlations between responses to the velogenic and lentogenic NDV strains. Results highlight possibilities for the selective breeding of local chicken ecotypes for improved resistance to NDV, with the goal of breeding local chickens that can perform favorably in conditions where NDV is endemic, and vaccination is problematic.

## 2. Materials and Methods

### 2.1. Experimental Design 

In Tanzania, 62 roosters and 302 hens from the Kuchi, Morogoro Medium (MoroMed), and Ching’wekwe (Ching) ecotypes that also produced experimental chickens for the lentogenic infection study described by [17] were used to breed chickens for the natural velogenic NDV exposure trials. Similarly, in Ghana, 42 roosters and 128 hens from the Coastal Savannah (CS), Forest (FO), and Interior Savannah (IS) ecotypes that also produced experimental chickens for the lentogenic study described by [18] were used. All breeder birds in the two countries were raised in a controlled environment with ad libitum access to both feed and water. Experimental birds were produced in 3 hatches for the Tanzania ecotypes and in 6 hatches for the Ghana ecotypes, with each hatch comprising one replicate of the natural exposure trial. For all replicates, chicks were raised in a single pen under similar conditions, with ad libitum access to both water and feed.

For each replicate of the natural challenge, birds that were suspected to be infected with velogenic NDV based on clinical signs were purchased from farmers or at a local market. Oropharyngeal and cloacal swabs samples were collected and RT-qPCR was performed to confirm the birds’ infection with mesogenic or velogenic NDV [29,30]. Sick birds were also screened for absence of infection with avian influenza virus, using a FluDETECT^®^ Avian kit (Zoetis, Parsippany-Troy Hills, NJ, USA), in order to avoid introducing another major respiratory virus into the study. For each replicate, the screened sick birds were mixed with 40 healthy naïve birds in an isolated pen to amplify the infection and to standardize timing. Three days later, upon development of clinical signs, the resulting seeder birds were mixed with healthy experimental birds in a ratio of 25:1 in Ghana and 20:1 in Tanzania. Experimental birds were 28 days of age at the time of exposure for all replicates in Ghana, and 42, 35, and 28 days of age for replicates 1, 2, and 3, respectively, in Tanzania. All trials in both countries were terminated at 21 days post-exposure (dpe) and surviving birds were euthanized at that time. Birds that died before the end of the experiment died before 21 days. Across replicates, the natural exposure trials were conducted on a total of 1365 chickens in Ghana (CS (340), FO (708), and IS (317)), and on 1556 chickens in Tanzania (Kuchi (439), Ching (405), and MoroMed (712)). 

### 2.2. Phenotypic Measurements 

The natural exposure to velogenic NDV caused severe clinical symptoms and allowed observation of phenotypic responses of birds’ post-exposure. Survival time was recorded as the number of dpe when the bird died; birds that survived until 21 dpe were euthanized. Viral RNA was isolated from lachrymal fluid (tears) and cloacal swabs at 2, 4, 6, and 10 dpe using a MagMAX-96 viral RNA isolation kit (Life Technologies, Carlsbad, CA, USA) and quantified using qRT-PCR, following the procedures described in [31]. Average viral RNA was computed and transformed to log_10_. Blood samples were collected at 7 dpe to measure anti-NDV antibody levels by ELISA (IDEXX, Westbrook, ME, USA). Blood samples were also collected from all birds before NDV exposure using an enzyme-linked immunosorbent assay to measure the natural anti-KLH antibody in birds (MP Biomedicals Inc., Aurora, OH, USA). Lesions in the trachea, proventriculus, intestine (duodenum, jejunum, and ileum), and cecal tonsils were assessed and scored on the dead and euthanized birds on a scale of 0 to 5, where 0 indicated no visible lesions and 5 indicated severe lesions. Lesions scores were averaged across organs for each bird. Body weights were recorded at hatch and weekly thereafter prior to exposure, and at 0, 2, 4, 6, and 10 dpe. Pre-exposure growth rate (g/d) was calculated based on body weights at hatch and at exposure. Post-exposure growth rate (g/d) was calculated based on body weights at 0 and 4 dpe because few birds survived past 4 dpe. 

### 2.3. Bird Genotyping and Quality Control

Whole blood was collected from breeders and chicks before exposure and placed on Whatman FTA cards (Sigma-Aldrich, St. Louis, MO, USA) for genomic DNA extraction. For breeders, genotyping was conducted on an Affymetrix Axiom 600K Array (Thermo Fisher Scientific Inc., Carlsbad, CA, USA) at GeneSeek (Lincoln, NE, USA). Genotypes for the Ghana and Tanzania breeders were combined, and genotype data quality control was performed using Axiom^TM^ Analysis Suite 3.1 (Applied Biosystems, Thermo Fisher Scientific Inc., Carlsbad, CA, USA), using similar thresholds as described in [18]. A total of 421,492 SNPs remained after quality control and were utilized for imputation analyses. The experimental birds were genotyped using targeted genotyping by sequencing (GBS) for the 5K SNP low-density panel described by [32], with 100 bp paired ends on 4 lanes of an Illumina NextSeq500 Hiseq at GeneSeek (Lincoln, NE, USA). Raw read sequence data were processed using an in-house shell script that utilized publicly available software tools (BWA (0.7.17), SAMtools (1.9), PICARDS (2.17.0), and BCFtools (1.9)), as described by [32]. Reads were aligned to the Gallus gallus version 5 reference genome and genotypes were called from the vcf using an in-house python script. The low density genotyped experimental birds were imputed to the high-density panel that the breeders were genotyped on using Fimpute [33], separately for each country. Experimental birds were assigned to half and full-sib families based on the distribution of genomic relationships among birds within each country. 

### 2.4. Data Analyses 

The three ecotypes within each country were previously found to have partial shared ancestries based on analysis of the high-density SNP genotypes of birds used in the lentogenic NDV challenge studies, as described by [17,18]. These studies used Admixture software [34] to determine the optimal number of subpopulations, which were 3 for the Ghana ecotypes [18] and 2 for the Tanzania ecotypes [17]. This same procedure was applied to the imputed genotypes of the experimental birds for the velogenic NDV exposure trials to quantify the proportional contributions of the 3 and 2 subpopulations for the Ghana and Tanzania experimental birds, respectively. 

Genetic parameters were estimated both by country and using the combined data. Asreml 4.2 software [35] was used to estimate variance components and heritability for each trait using the following univariate animal model: (1)$$Y_{ijkl}=µ+R_{i}+P_{j}+A_{k}+M_{l}+e_{ijkl}$$
where *Y* is the phenotype, i.e., pre- and post-exposure growth rate, trachea, proventriculus, intestinal, cecal tonsil, and average lesion scores, natural antibody level, NDV antibody level at 10 dpe, tear and cloacal viral loads at 2, 4, 6, and 10 dpe, and survival time. Fixed effects included replicate, *R* (1 to 6 for Ghana and 1 to 3 for Tanzania), and population proportions (P), obtained as described above, fitted as one covariate for Tanzania and as two covariates for Ghana, as described by [17,18]. Random effects included animal genetic effects (*A*) with a genomic relationship matrix obtained based on the procedures described by [36], maternal environmental effects (*M*), and residuals (*e*). The maternal effect was removed from the model for traits for which it was estimated to be zero. For viral load traits and for natural and anti-NDV antibody levels, the effect of assay plate was added to the model as a fixed effect. For lesion scores, the effect of bird survival to the end of the trial (0/1) was added as a fixed effect. For the combined analyses, replicate by country and country-specific population proportion covariates were added. Phenotypic variance was estimated as the sum of the estimates of the animal genetic, maternal (if fitted), and residual variances. Heritability was estimated as the ratio of animal genetic and phenotypic variance estimates.

Phenotypic and genetic correlations between traits recorded in the velogenic NDV trials were estimated separately for each country and combined using a bivariate version of the univariate models described above. Genetic correlations of the velogenic NDV phenotypes of the current study with phenotypes recorded in the lentogenic NDV trials [17,18] (pre- and post-infection growth rate, anti-NDV antibody levels at 10 days post-infection (dpi), viral load at 2 and 6 dpi, and viral clearance) were also estimated. Bivariate models were used, with the model as described above, for the velogenic NDV response traits and as described in [17,18] for the lentogenic NDV response traits. Because the lentogenic and velogenic NDV challenges were conducted using different birds but from the same ecotypes, environmental correlations were not estimable and were set equal to zero. 

## 3. Results

### 3.1. Survival Time and Lesion Scores

The velogenic NDV natural exposure caused high mortality, with average survival times of 7 and 9 dpe for Tanzania and Ghana, respectively. Across all replicates, only 11 and 89 birds survived to the end of the experiment for Tanzania and Ghana, respectively. Mortality varied across the six Ghana trial replicates, with up to 100% mortality for replicates 4, 5, and 6, respectively, and the lowest mortality of 73% in replicate 2 (Figure 1). Over 98% mortality was observed in the Tanzania study across the three replicates. For both Tanzania and Ghana, birds that survived to 21 dpe were euthanized and, hence, their survival time was set to 21 days. The average survival time of approximately 7 days was the same across the three replicates for Tanzania, but varied across the six replicates for Ghana, with higher average survival times of 16 and 9 days for replicates 2 and 3, respectively. Replicates 4 and 6 had the lowest average survival time of 5 days. Average lesion scores were higher for birds that died than for birds that survived to 21 dpe in both Ghana and Tanzania (Figure 2). Although average lesion scores were less than 1.5 for birds that died or survived across the six (Ghana) and three (Tanzania) replicates, birds that died had higher average lesion scores in replicate 1 and 2 for Ghana and Tanzania, respectively (Figure 3 and Appendix A).

### 3.2. Subpopulation Proportions and Effects 

There was clear evidence of admixture between ecotypes, with overlap among the three ecotypes within each country (Figure 4). There were three and two distinct subpopulations for the ecotypes from Ghana and Tanzania, respectively, with very few birds comprising 100% of one subpopulation. We observed significant differences in the effect of population proportions for pre-exposure growth rate, tear viral load at 6 dpe, cloacal viral load at 4 dpe, and survival time for Ghana, for which two subpopulation proportions were fitted in the model (Table 1). 

### 3.3. Heritability Estimates 

Heritability estimates based on univariate analyses of data from each country and combined are shown in Table 2. Heritability estimates from the bivariate trait analyses (not shown) were similar to those of the univariate analyses. For Tanzania, low to moderate estimates of heritability, ranging from 0.14 to 0.32, were observed for average lesions scores, natural antibody, survival time, proventriculus lesions scores, and pre-exposure growth rate. Estimates of heritability were in the same range for these traits for Ghana, except for natural antibody level, which was not heritable for Ghana. Although anti-NDV antibody level at 7 dpe was not heritable in Tanzania, it was heritable in Ghana, with an estimate of 0.21. Survival time was estimated to be moderately heritable at 0.23 in Tanzania and 0.27 in Ghana. Although sample sizes were small for the viral load traits, our results showed a moderate heritability estimate for cloacal viral load at 4 and 6 dpe for Ghana. For the combined country data analyses, estimates of heritability were low to moderate for tear (6 dpe) and cloacal (2, 4, and 6 dpe) viral load, natural antibody, anti-NDV antibody levels (7 dpe), lesions scores (proventriculus and average lesions), pre-exposure growth rate, and survival time (0.21), ranging from 0.10 for average lesions to 0.35 for pre-exposure growth rate. Trachea lesion scores and tear viral load at 2 dpe were not heritable. 

### 3.4. Correlations among Velogenic NDV Response Traits

Phenotypic correlations among traits were mostly low, for both the individual and combined country data analyses (Table 3). However, survival time was positively correlated with anti-NDV antibody levels at 7 dpe in the combined data analyses, and lesion score traits were relatively highly correlated with each other. Correlations were not computed for some cloacal and tear viral load traits for individual countries because of their low sample sizes. 

For the individual country data analyses, survival time was negatively genetically correlated with both proventriculus and intestinal lesion scores for both Tanzania and Ghana (Table 3). As expected, lesion score traits (proventriculus, intestinal, and average lesion score traits) were positively correlated with each other, and there was a high negative genetic correlation between lesion score traits and anti-NDV antibody levels at 7 dpe for both countries. Natural anti-KLH antibody level was positively correlated with proventriculus and average lesion scores, and negatively correlated with cecal and intestinal lesion scores. 

Estimates of genetic correlations for the combined analyses generally followed a similar trend as the individual country estimates (Table 3). Survival time was positively correlated (genetically) with pre- and post-exposure growth rate, negatively correlated with cloacal viral load at 2 dpe and tear viral load at 6 dpe. Moreover, survival time was also negatively correlated with all analyzed lesion score traits (proventriculus, intestinal, cecal tonsil, and average lesions) and positively correlated with anti-NDV antibody level at 7 dpe. Average lesion score was negatively correlated with cloacal viral load at 2 dpe and positively correlated with cloacal viral load at 4 dpe. Like individual country analyses, anti-NDV antibody level at 7 dpe was negatively correlated with the analyzed lesion score traits (proventriculus, intestinal, cecal tonsil, and average lesion score). Pre-exposure growth rate was positively correlated with lesion score traits, with anti-NDV antibody level at 7 dpe, and with survival time, and was negatively correlated with cloacal viral load at 2 and 4 dpe. Natural antibody levels were positively correlated with anti-NDV antibody at 7 dpe and with cloacal viral load at 2 and 4 dpe. 

In a separate analysis, we analyzed lesion scores for Tanzania and fitted survival time as a linear or quadratic effect in the models. Estimates of genetic correlations between lesion score traits and other NDV response traits were zero or reduced, except for correlations among the adjusted lesion score traits. This approach was not explored further for the combined data analyses. 

### 3.5. Genetic Correlations between Velogenic and Lentogenic NDV Response Traits

Correlation estimates of genetic correlations of velogenic with lentogenic NDV response traits are reported in Table 4. All these estimates had very high standard errors. 

## 4. Discussion

To the best of our knowledge, this study is the first to report genetic parameters for response of local African chicken ecotypes following natural contact exposure to velogenic NDV. This novel natural challenge approach was used to mimic rural farms or backyards, where NDV is endemic, in order to maximize expression of genetic differences in resilience of unvaccinated local chickens to velogenic NDV. It should be noted that each replicate may have been exposed to different and potentially multiple NDV strains, because birds with ND symptoms were obtained from live bird markets at different times for each replicate and used to infect seeder birds. Apart from avian flu, which the sick birds were screened for, we can also not rule out the presence of co-infecting pathogens in the environment and, if present, they may also have varied between replicates. However, these sources of variation between replicates were an important part of the experimental design in order to estimate genetic parameters that were robust over a variety of NDV strains, co-infecting pathogens, and environments. This is in contrast to typical experimental infection studies, which by design employ rather than narrowly define conditions. While providing a “cleaner” experimental design, these studies do not yield information of broad application in the field. The natural exposure approach allowed the collection of various phenotypes, including growth rate, survival time, lesion scores of various organs, antibody levels in blood, and viral load in tears and cloacal swabs.

Admixture analyses indicated that birds from the three ecotypes within each country shared similar genetic backgrounds. This agrees with previous studies of these populations [17,18] and is likely the result of inter-breeding among ecotypes and movement of birds through market chains and trade. 

Although we generally observed similar average survival times across the three replicates in Tanzania, average survival times differed between replicates in Ghana. The latter could be due to the difference in NDV strains to which the birds were exposed. 

A previous study grouped NDV isolates collected in several African countries according to the updated NDV classification system [37] and identified up to three genotypes in Tanzania (V, VII, and XIII) [13,14,38] and one genotype in Ghana (XVIII) [13,14]. However, within each genotype class consists a collection of NDV strains and it is likely that, even in Ghana, different strains were present in the different replicates. The different NDV strains and varying environmental conditions likely contributed to the differences in survival time between the two countries and between replicates within each country. The different NDV genotypes and varying environmental conditions could also have contributed to the differences in survival times between the two countries. Although the presence of different NDV strains [13] across replicates is not ideal from an experimental perspective, it is desirable from a practical perspective, as the goal is to select birds that are less susceptible to multiple strains including both meso- and velo-genic NDVs. To gain additional insight in the strains that were present in each replicate, we are currently studying the genotypes of NDV viruses that circulated in each replicate. 

Infection with velogenic NDV is known to cause lesions in different tissues (trachea, proventriculus, intestine, and cecal tonsil) [39]. In our study, birds that survived had lower average lesion scores in both Ghana and Tanzania, but lesion scores were generally not severe across the four tissues in either the dead or surviving birds. Thus, most mortality was likely not caused by tissue lesions. In addition, most mortalities occurred before 9 dpe for both Tanzania and Ghana. Mortality from velogenic NDV can be caused by mechanisms other than tissue lesions, such as atrophy and depletion of lymphocytes in the lymphoid organs [15,40]. We did not analyze mortality as a binary trait in the current study because of the very low number of surviving birds within each country, especially for Tanzania. Instead, survival time was evaluated as the main indicator of resilience.

A previous study identified a significant difference in the severity of tissue (proventriculus, intestinal, and cecal tonsil) lesions in 10-week-old chickens (broilers and pullets) that were infected with velogenic NDV, with very severe lesions observed in broilers compared to pullets [20]. The differences in NDV susceptibility between breeds or ecotypes might explain differences in clinical signs and mortality between Ghana and Tanzania local chicken ecotypes.

Estimates of heritability for pre-exposure growth rate were 0.32 ± 0.06 and 0.37 ± 0.06 for Ghana and Tanzania, respectively. These estimates are similar to those in the lentogenic NDV trials that were conducted using birds from these same populations, where birds were infected with the LaSota NDV vaccine strain at 28 days of age (doa) in Ghana [18] and Tanzania [17]. Heritability estimates for survival in the current study ranged from 0.21 to 0.27 for the Ghana, Tanzania, and combined data analyses. These estimates are substantially higher than the estimate of 0.09 obtained in a recent study conducted on local chicken ecotypes in Cameroon [41]. Only proventriculus lesion scores had heritability greater than 0.10 for the Ghana, Tanzania, and combined data analyses among the lesion score traits. Although lesion development is one of the symptoms observed upon NDV challenge, it is important to note that some birds died immediately or a day after challenge with limited tissue lesion development. More studies are needed to understand the genetic basis of lesion development in local chicken ecotypes after NDV challenge/infection. The estimate of heritability for natural anti-KLH antibody was greater than 0.10 for the Ghana, Tanzania, and combined data, but anti-NDV antibody at 7 dpe was only heritable in the Ghana and the combined data analyses. Natural anti-KLH antibody response has also been reported to be heritable in laying chickens [42] and could improve survival through enhanced disease resistance [43]. The current study had low sample for anti-NDV antibody levels because of mortality before sample collection for the antibody assay, and a larger population size might be needed to better reflect the polygenetic mode of action of genes for anti-NDV antibody response. Viral load traits were only heritable for the Ghana and the combined data analyses. Viral load samples were also collected in the relatively later stages of the trials and, hence, sample sizes were low. Further trials are necessary with larger sample sizes to provide a better understanding of tear and cloacal viral load upon velogenic NDV challenge. 

Overall, the heritability estimates were mostly low for most traits, and moderate for pre-exposure growth rate and survival time in the combined data, ranging from 0.11 for average lesions scores to 0.36 for pre-exposure growth rate. The moderate heritability estimates suggest that natural anti-KLH antibody and velogenic NDV response traits, such as survival time, could be improved using selective breeding to enhance chicken NDV resilience. Heritability estimates were low for almost all immune and disease-related traits. Notably, however, estimates for these traits tend to be low even in well-controlled NDV studies conducted in ABSL-2 rooms, with controlled temperature and lighting, consistent high-quality feed, one genetic line, and a single stock of virus [31].

Phenotypic correlations among recorded traits were mostly low, except for correlations among lesion score traits. The high phenotypic correlations between tissue lesion score traits and average lesion scores are not surprising because average lesion scores were derived from the tissue lesion scores. The phenotypic correlations between survival time and lesion score traits were low and negative, with correlations of −0.11 and −0.10 observed for cecal tonsil and average lesion scores, respectively, in the combined data. Although we would expect the correlations between tear and cloacal swabs to be substantial, these correlations were low in the current study, but standard errors were sizable. 

Survival time was also negatively genetically correlated with all the lesion score (proventriculus, intestinal, cecal tonsil, and average lesions) traits for Tanzania, Ghana, and combined data estimates. Selection for increased survival time could reduce gastro-intestinal tract (GIT) lesion development, which may enable birds to ultimately fight off velogenic NDV. A previous study indicated that NDV infection caused more mortalities and higher lesion scores in broilers than in pullets [20]. In chickens, high GIT lesion infestation is often attributed to hemorrhages and ulcers, and this can significantly increase the severity of NDV infection in chickens compared to other poultry species [19]. In addition, the negative genetic correlation estimates of survival time with cloacal viral load at 4 dpe (−0.55) for the Ghana data, with similar trends for cloacal viral load at 2 dpe (−0.82) and tear viral load at 6 dpe (−0.59) for the combined data, suggests that selection for survival time might reduce viral load against virulent NDV challenge. Note, however, that the sample size for the viral load traits was low in our study and larger sample sizes are needed to draw better inferences. 

Survival against virulent NDV infection is a complex trait and is controlled by many genes, however, it is beneficial to select for increasing survival time even if birds eventually die because each gene involved might be inducing a different survival mechanism [44]. It is also important to note that we ignored censored data for survival in the current study, and that we assumed that lesion scores in birds that survived and died were the same trait. Birds that survived up to 21 doa were euthanized and a postmortem was performed to score lesions in the various GIT tissues. 

The negative genetic correlation estimates in the Ghana data of anti-NDV antibody levels at 7 dpe with lesion score traits (>−0.60), as well as its positive genetic correlation estimate with natural antibody levels (0.90 ± 0.87) suggest that selection for birds with increased natural anti-KLH antibody production may enable birds to respond with increased anti-NDV antibody production upon velogenic NDV infection, ultimately reducing tissue lesion development. Although these genetic correlation estimates are moderate to high, it is important to consider some of these estimates with caution because of their high standard errors. 

Standard errors of estimates of genetic correlations between the lentogenic (vaccine) and naïve (velogenic) studies were very high. Therefore, it is not possible to draw inference from the reported results on the ability to use response to the vaccine strain to select for improved response to velogenic NDV. 

## 5. Conclusions

This study is the first to analyze a substantial number of African local chicken eco-types following “natural contact NDV exposure”, and provides a very useful protocol and baseline for future studies into genetic resistance of chickens to velogenic NDV. Our results indicate response to natural exposure to velogenic NDV is heritable for both the Ghana and Tanzania ecotypes. Considering the moderately high heritability estimates for survival time and the trends of favorable genetic correlations of survival time with tissue lesion scores, growth rate, and viral load traits, we provide evidence that virulent NDV response traits can be improved in local Ghanian and Tanzanian chicken ecotypes through selective breeding for increased survival time following natural exposure to velogenic NDV. The production of local chicken ecotypes that can perform in regions where NDV is endemic would be crucial in enhancing household food security. 

## Figures and Tables

**Figure 1 animals-12-02755-f001:**
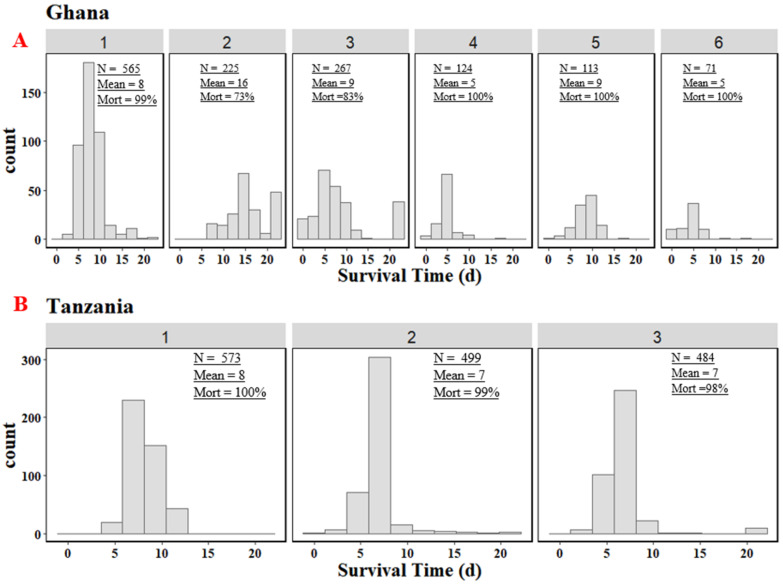
Survival time in days for 6 replicates for Ghana (**A**) and 3 replicates for Tanzania (**B**). N = number of birds in the replicate; Mean = average survival from exposure to 21 days within the replicate; Mort = % of birds that survived to the end of the experiment (21 days after exposure).

**Figure 2 animals-12-02755-f002:**
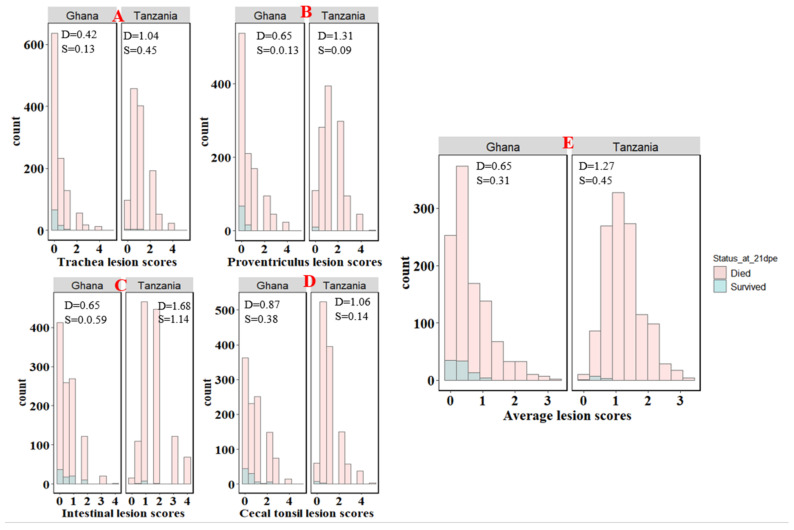
Distributions of lesions scores in (**A**) trachea, (**B**) proventriculus, (**C**) intestine, (**D**) cecal tonsil, and (**E**) average across the four tissues for birds that died and survived to 21 days post exposure (dpe). Numbers in the plots represent the mean scores for the birds that died (D) and survived (S).

**Figure 3 animals-12-02755-f003:**
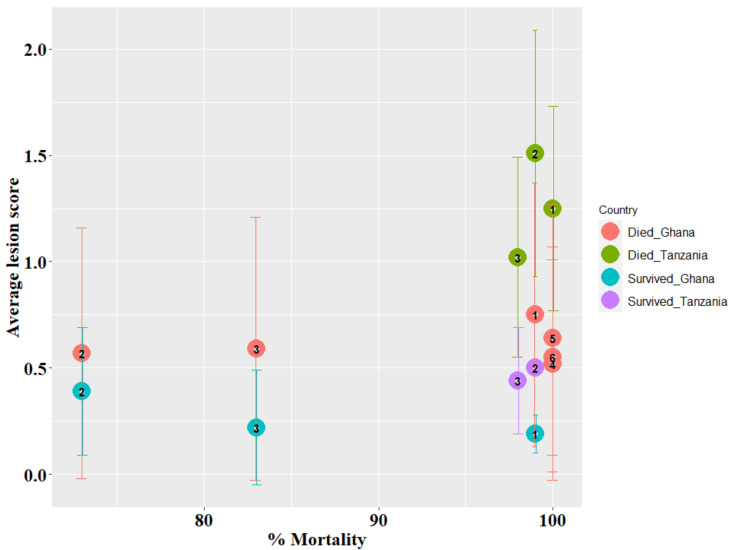
Average lesion scores and standard deviation bars for birds that died and survived for each replicate in Ghana and Tanzania, as identified by the colors and numbers, plotted against the % mortality in the corresponding replicate.

**Figure 4 animals-12-02755-f004:**
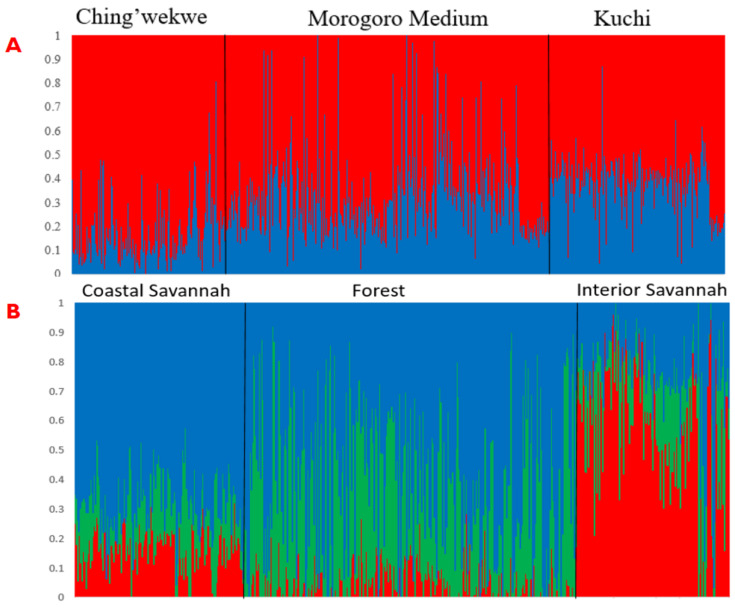
Admixture plot showing mixed ancestry among birds from the three Tanzanian ecotypes (**A**) (Ching’wekwe, Morogoro Medium, and Kuchi) and Ghanian ecotypes (**B**) (Coastal Savannah, Forest, and Interior Savannah).

**Table 1 animals-12-02755-t001:** Estimates of the effect of subpopulation proportions (s.e.).

	Ghana Subpopulations		Tanzania
Trait	Red	Green	Red
Pre-exposure growth rate (g/d)	0.61 (0.29)	0.35 (0.26)	1.22 (0.53)
Pre-exposure growth rate (g/d)	0.96 (0.62)	1.15 (0.59)	−2.02 (1.77)
Trachea lesion score	0.16 (0.09)	−0.04 (0.09)	0.39 (0.11)
Proventriculus lesion score	0.08 (0.19)	−0.15 (0.17)	0.33 (0.34)
Intestinal lesion score	0.08 (0.10)	0.05 (0.10)	0.21 (0.21)
Cecal tonsil lesion score	0.17 (0.17)	−0.07 (0.16)	0.22 (0.18)
Average lesion score	0.09 (0.10)	−0.05 (0.10)	0.28 (0.144)
Natural antibody ^1^	−0.06 (0.07)	0.04 (0.07)	−0.09 (0.14)
Anti-NDV antibody 7 dpe ^1^	0.03 (0.18)	0.24 (0.17)	−0.24 (0.18)
Survival Time (days)	0.82 (0.99)	2.43 (0.90)	−0.51 (0.55)
Tear viral load 2 dpe ^1^	−0.06 (0.15)	−0.06 (0.15)	0.03 (0.08)
Tear viral load 6 dpe ^1^	0.48 (0.51)	−0.05 (0.48)	0.10 (0.24)
Cloacal viral load 2 dpe ^1^	0.06 (0.23)	−0.01 (0.21)	0.18 (0.13)
Cloacal viral load 4 dpe ^1^	0.02 (0.23)	0.25 (0.21)	0.12 (0.18)

Subpopulation names refer to the colors in Figure 1. The effect of the blue subpopulations was set equal to zero for Ghana and Tanzania. ^1^ Log_10_ transformation.

**Table 2 animals-12-02755-t002:** Phenotypic descriptive statistics and estimates of variance components (proportions of phenotypic variance).

	Tanzania				Ghana				Combined			
	N ^1^	Mean (SD)	h ^2^ (se)	Dam (se)	N ^1^	Mean (SD)	h ^2^ (se)	Dam (se)	N ^1^	Mean (SD)	h ^2^ (se)	Dam (se)
Pre-exposure growth (g/d)	1239	3.82 (1.56)	0.32 (0.06)	0.02 (0.00)	1344	4.34 (1.68)	0.37 (0.06)	0.00 (0.01)	2572	4.16 (1.61)	0.35 (0.04)	0.01 (0.01)
Post-exposure growth (g/d)	1002	2.93 (5.33)	0.01 (0.05)	0.03 (0.03)	816	0.14 (3.93)	0.01 (0.07)	0.04 (0.03)	1818	1.69 (8.42)	0.04 (0.04)	0.03 (0.02)
Trachea lesions	1239	1.03 (0.83)	0.00 (0.00)	0.04 (0.01)	1170	0.43 (0.77)	0.00 (0.04)	-	2409	0.72 (0.83)	0.01 (0.03)	0.01 (0.01)
Proventriculus lesions	1239	1.29 (0.98)	0.23 (0.07)	0.04 (0.02)	1170	0.61 (0.90)	0.14 (0.06)	0.01 (0.02)	2409	0.96 (1.01)	0.19 (0.05)	0.02 (0.02)
Intestinal lesions	1239	1.66 (0.91)	0.09 (0.04)	-	1170	0.66 (0.77)	0.03 (0.04)	-	2409	1.18 (0.97)	0.08 (0.03)	-
Cecal Tonsil lesions	1239	1.04 (0.87)	0.05 (0.04)	-	1170	0.84 (0.94)	0.09 (0.05)	-	2409	0.95 (0.92)	0.08 (0.03)	-
Average lesion score	1239	1.26 (0.55)	0.14 (0.06)	0.02	1170	0.64 (0.62)	0.08 (0.05)	-	2409	0.95 (0.66)	0.10 (0.03)	0.01 (0.03)
Natural antibody ^3^	430	7.11 (1.92)	0.18 (0.14)	0.05	936	3.70 (0.45)	0.11 (0.10)	0.01 (0.02)	1960	4.35 (0.79)	0.13 (0.06)	0.04 (0.02)
Anti-NDV antibody 7 dpe ^3^	298	1.98 (0.65)	0.00 (0.00)	0.00	576	2.03 (0.72)	0.21 (0.10)	-	874	2.85 (1.87)	0.12 (0.08)	-
Survival_time (days)	1238	7.12 (2.10)	0.23 (0.06)	0.00	1160	9.11 (5.19)	0.27 (0.06)	0.00	2398	8.00 (3.97)	0.21 (0.04)	-
Tears viral load 2 dpe ^3^	660	0.11 (0.39)	-	-	546	1.36 (1.06)	0.00 (0.00)	-	617	1.23 (1.07)	0.00 (0.00)	-
Tears viral load 6 dpe ^3^	266	1.15 (1.42)	-	0.9 (0.06)	187	3.59 (1.47)	0.44 (0.29)	-	454	3.16 (1.56)	0.28 (0.15)	-
Cloacal viral load 2 dpe ^3^	280	0.44 (0.57)	0.06 (0.19)	-	546	1.78 (0.33)	0.00 (0.00)	0.05 (0.04)	826	0.28 (0.51)	0.16 (0.11)	-
Cloacal viral load 4 dpe ^3^	297	0.56 (0.69)	0.00 (0.00)	-	428	3.19 (0.81)	0.38 (0.14)	0.04 (0.04)	725	0.59 (0.74)	0.21 (0.11)	0.03 (0.04)
Cloacal viral load 6 dpe ^3^	143	-	-	-	310	4.51 (1.17)	0.17 (0.17)	0.12 (0.07)	453	1.05 (1.07)	0.15 (0.15)	0.06 (0.06)

^1^ Number of phenotypic records, ^3^ Log_10_ transformation, SD = Standard deviation, se = Standard error. h ^2^ = heritability.

**Table 3 animals-12-02755-t003:** Estimates (s.e) of phenotypic (lower diagonal) and genetic (upper diagonal) correlations analyses for the combined data (first row), Tanzania (second row), and Ghana (third row).

	Growth Rate		Lesion Scores				Antibody		Survival Time	Viral Load		
	Pre- Exposure	Post-Exposure	Proventri-culus	Intestine	Cecal Tonsil	Average	Natural^2^	7 dpe^2^		Cloacal 2 dpe^2^	Cloacal 4 dpe^2^	Tears 6 dpe^2^
Growth rate pre-exposure ^1^		- −0.08 (0.27) −0.22 (0.31)	0.09 (0.11) 0.14 (0.15) 0.01 (0.19)	0.56 (0.16) 0.55 (0.21) 0.56 (0.39)	0.39 (0.17) 0.60 (0.34) 0.25 (0.22)	0.40 (0.14) 0.47 (0.17) 0.23 (0.22)	0.25 (0.16) 0.09 (0.25) −0.18 (0.05)	0.77 (0.88) - −0.10 (0.23)	0.12(0.11) −0.14 (0.19) 0.21 (0.15)	−0.27 (0.21) - -	−0.27 (0.21) - -	- - -
Growth rate post-exposure ^1^	0.01 (0.04) −0.04 (0.03) −0.11 (0.03)		0.23 (0.22) - -	−0.26 (0.29) - -	−0.25 (0.29) - -	0.07 (0.26) - -	0.10 (0.25) - -	0.07 (0.40) - -	0.48 (0.65) - -	0.81 (0.54) - -	0.01 (0.36) - -	−0.28 (0.42) - -
Proventri-culus lesion score	0.06 (0.02) 0.05 (0.03) 0.08 (0.03)	0.04 (0.02) 0.06 (0.03) 0.10 (0.04)		- −0.43 (0.26) 0.95 (0.44)	0.43 (0.17) 0.21 (0.03) 0.74 (0.27)	0.86 (0.06) 0.84 (0.09) 0.88 (0.15)	0.23 (0.16) 0.11 (0.39) 0.11 (0.45)	−0.66 (0.21) - −0.93 (0.24)	−0.23 (0.13) −0.17 (0.22) −0.33 (0.31)	−0.14 (0.25) - -	0.05 (0.22) - 0.29 (0.30)	−0.27 (0.25) - -
Intestine lesion score	0.11 (0.02) 0.11 (0.03) 0.09 (0.03)	−0.01 (0.02) −0.01 (0.03) 0.06 (0.04)	0.11 (0.02) 0.15 (0.03) 0.46 (0.02)		0.91 (0.23) 0.23 (0.03) -	0.81 (0.13) 0.83 (0.18) -	0.03 (0.24) −0.33 (0.42) -	−0.28 (0.37) - -	−0.25 (0.20) −0.31 (0.36) -	−0.20 (0.39) - -	0.44 (0.32) - -	−0.48 (0.37) - -
Cecal tonsil lesion score	0.06 (0.02) 0.04 (0.34) 0.10 (0.34)	−0.02 (0.02) −0.01 (0.02) 0.04 (0.04)	0.40 (0.02) 0.21 (0.04) 0.57 (0.02)	0.37 (0.02) 0.23 (0.03) 0.57 (0.02)		0.80 (0.10) −0.70 (0.38) 0.85 (0.14)	0.07 (0.23) −0.01 (0.05) −0.53 (0.60)	−0.53 (0.34) - −0.68 (0.30)	−0.38 (0.20) −0.10 (0.03) −0.14 (0.35)	−0.48 (0.36) - -	0.40 (0.31) - 0.32 (0.32)	0.81 (0.49) - -
Average Lesion score	0.13 (0.02) 0.13 (0.03) 0.13 (0.03)	0.01 (0.02) 0.03 (0.03) 0.06 (0.04)	0.72 (0.01) 0.65 (0.02) 0.78 (0.01)	0.65 (0.01) 0.61 (0.02) 0.74 (0.01)	0.75 (0.01) 0.62 (0.02) 0.85 (0.01)		0.22 (0.21) 0.26 (0.43) 0.81 (0.58)	−0.56 (0.28) - −0.68 (0.30)	−0.40 (0.17) - -	−0.37 (0.30) - -	0.42 (0.27) - 0.62 (0.35)	0.07 (0.32) - -
Natural Antibody^2^	0.11 (0.03) 0.29 (0.06) −0.07 (0.06)	0.05 (0.03) 0.10 (0.06) 0.07 (0.05)	0.02 (0.02) −0.05 (0.07) 0.00 (0.04)	−0.01 (0.02) −0.05 (0.05) 0.00 (0.04)	−0.03 (0.02) 0.13 (0.06) −0.07 (0.05)	−0.01 (0.02) −0.05 (0.06) −0.06 (0.05)		0.34 (0.30) - 0.90 (0.87)	−0.04 (0.18) −0.05 (0.40) −0.09 (0.34)	0.43 (0.36) - -	0.14 (0.24) - -	−0.01 (0.32) 0.13 (0.65) -
Antibody 7 dpe^2^	0.05 (0.03) - −0.07 (0.06)	−0.09 (0.04) - −0.20 (0.05)	−0.12 (0.04) - −0.17 (0.05)	−0.04 (0.04) - −0.05 (0.05)	−0.07 (0.04) - 0.04 (0.09)	−0.09 (0.04) - −0.11 (0.05)	0.09 (0.04) - -		0.48 (0.88) - 0.04 (0.27)	0.46 (0.42) - -	0.21 (0.34) - 0.39 (0.31)	−0.49 (0.47) - -
Survival time	0.15 (0.02) 0.06 (0.03) 0.21 (0.03)	0.05 (0.03) 0.08 (0.03) 0.23 (0.04)	−0.08 (0.02) 0.03 (0.03) −0.06 (0.03)	−0.03 (0.02) −0.06 (0.03) −0.03 (0.03)	−0.11 (0.02) −0.09 (0.03) −0.04 (0.03)	−0.10 (0.02) −0.01 (0.03) −0.01 (0.03)	0.03 (0.03) 0.18 (0.04) 0.02 (0.04)	0.17 (0.04) - 0.06 (0.04)		−0.82 (0.59) - -	0.04 (0.22) - −0.55 (0.31)	−0.59 (0.58) - -
Cloacal viral load 2 dpe^2^	−0.01 (0.04) - -	−0.06 (0.06) - -	−0.06 (0.04) - -	−0.02 (0.04) - -	−0.02 (0.04) - -	−0.04 (0.04) - -	0.01 (0.04) - -	0.01 (0.06) - -	−0.15 (0.04) - -		0.78 (0.39) - -	−0.12 (0.46) - -
Cloacal viral load 4 dpe^2^	−0.01 (0.04) - -	−0.41 (0.05) - -	−0.01 (0.04) - 0.03 (0.05)	−0.04 (0.04) - -	0.04 (0.04) - 0.07 (0.05)	0.03 (0.04) - 0.08 (0.05)	0.01 (0.05) - -	0.10 (0.07) - 0.18 (0.09)	0.02 (0.03) - -	0.04 (0.05) - -		0.43 (0.37) - -
Tears viral load 6 dpi ^2^	- 0.01 (0.10) -	−0.01 (0.06) 0.02 (0.09) -	0.04 (0.05) 0.02 (0.09) -	0.01 (0.05) 0.06 (0.09) -	0.11 (0.05) 0.04 (0.09) -	0.08 (0.05) - -	0.06 (0.06) −0.39 (0.58) -	−0.01 (0.05) - -	−0.13 (0.04) - -	0.02 (0.07) - -	0.29 (0.08) - -	

^1^ Growth rate, ^2^ Log transformed.

**Table 4 animals-12-02755-t004:** Estimates of genetic correlations between traits measured in the Lasota and velogenic trials for Tanzania (top row) and Ghana (second row).

	Velogenic Trial Phenotypes						
		Growth rate pre-exposure	Proventriculus lesions	Intestine lesions	Average lesions	Natural antibody ^1^	Survival time
Lasota trialphenotypes	Growth rate pre-infection	0.64 (0.72) −0.34 (0.39)	0.85 (0.88) −0.05 (0.55)	0.13 (1.31) -	0.10 (0.83) −0.18 (0.71)	0.91 (1.27) −0.58 (1.10)	−0.04 (1.36) −0.13 (0.67)
	Growth rate pre-infection	0.14 (1.03) −0.48 (0.48)	0.76 (0.92) 0.56 (0.67)	−0.49 (1.38) -	0.004 (0.87) −0.14 (0.88)	−0.84 (2.08) −0.32 (1.26)	0.24 (1.44) −0.65 (0.78)
	Anti-NDV antibody ^1^	0.98 (1.29) −0.02 (0.60)	−0.30 (1.07) −0.69 (0.82)	−0.80 (1.70) -	−0.42 (1.07) 0.43 (1.11)	−0.01 (1.76) −0.19 (1.35)	−0.24 (3.18) 0.50 (1.01)
	Viral load, 2 dpi ^1^	0.44 (1.64) 0.03 (0.57)	−0.02 (1.41) 0.55 (0.81)	−0.08 (2.23) -	−0.02 (1.40) 0.90 (1.06)	−0.90 (2.22) 0.56 (0.98)	−0.07 (2.25) 0.16 (0.96)
	Viral load, 6 dpi ^1^	−0.64 (1.15) −0.15 (0.68)	−0.21 (0.99) 0.71 (1.05)	−0.88 (1.64) -	−0.65 (1.00) 0.39 (1.33)	0.80 (1.61) 0.77 (1.22)	0.73 (4.15) −0.89 (1.26)
	Viral clearance	- −0.06 (0.57)	- −0.28 (0.81)	- -	- −0.50 (1.05)	- 0.16 (1.32)	- 0.69 (1.01)

^1^ Log transformed; dpi = days post infection.

## Data Availability

The datasets generated for this study will be made available under EBI (Project: PRJEB39468).

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
