# Peer review of "Genetic Analyses of Response of Local Ghanaian Tanzanian Chicken Ecotypes to a Natural Challenge with Velogenic Newcastle Disease Virus"

_animals, 2022, doi:10.3390/ani12202755_

Round 1

Reviewer 1 Report

In same place there are a lot of repeatations.

Author Response

Manuscript has been revised to reduce repetition 

Reviewer 2 Report

Manuscript entitled “Genetic analyses of response of local and Ghanaian Tanzanian chicken ecotypes to a natural challenge with velogenic Newcastle disease virus” by Muhammed Walugembe and others analyze the outcomes of NDV infection on chickens from different ecotypes with the purpose of obtaining heritability estimates. The authors find that key traits are either not heritable or of low heritability, however they find some positive and negative correlations of the outcome with levels of infection, in agreement with previous knowledge of expected outcomes for NDV. The overall results (for example positive and negative correlations) are not revealing of any new pattern on NDV outcomes, and the low heritability observed may be explained by poor experimental design. Overall, it appears that too many experimental variables were allowed preventing the achievement of any meaningful conclusion. Surprisingly despite the weakness of the heritability evidence, the author concludes that “the response to mesogenic or velogenic NDV of these local chicken ecotypes can be improved by selective breeding”.

The heritability scores, including those obtained for lesions (0.11) were weak or moderate perhaps reflecting the experimental limitations of the study. Normally high heritability values of 0.5 mean that on average half of the differences among phenotypes of animals are genetic. Low values of approximately 0.1 mean that most of the differences are not genetic. For Tanzania, low to moderate estimates were found for natural antibody, survival time, proventriculus lesions scores, and pre-exposure growth rate. Estimates of heritability were in the same range for these traits for Ghana, except for natural antibody level, which was not heritable for Ghana. Heritability estimates for survival in the current study were greater than 0.10 for Ghana, Tanzania, and combined data analyses. Only proventriculus lesion score had heritability greater than 0.10 for Ghana, Tanzania, and combined data analyses among the lesion score traits.

As the authors clearly state in the introduction, the determination of heritability for disease response traits is difficult due to the “potential uncontrolled” nature of infections among individual animals. A previous study by the same group described very low heritability of NDV response traits after infection with low pathogenic NDV strains in similar type of experiments [references 18] [19] [26]. In the study reported here, “natural exposure” trials were conducted with chickens from local ecotypes from Tanzania and Ghana as previously evaluated in [17], [18], and [28].  Surprisingly, no efforts were made here to eliminate or to reduce some known important factors that may lead to “potentially uncontrolled” outcomes. In fact, additional “uncontrolable” factors were introduced by the use of a challenge method “by contact” and by the use of a poorly characterized inoculum. The team appear to have ignored the fact that many genotypes of NDV viruses circulate in Africa simultaneously at one location, that these viruses have often high genetic and antigenic variability, that these viruses have low, medium and highly virulence, as no effort was made to purify or to genetically characterize the inoculum. Several previous studies have indicated the existence of multiple bacterial, fungal and viral agents co-infecting poultry worldwide and this type of infectome is likely to be present in the inoculum and on the non-SPF birds used during the “natural” infectious process. The chicken infectome in Africa is not simply composed of avian influenza (the only other agent tested) but of a plethora of RNA and DNA viruses that are known to have immune suppressive effect (e.g., Gumboro), synergistic respiratory effects (e.g. Infectious Bronchitis), and opportunistic disease causing effects (e.g. bacteria in digestive, respiratory and immune tissues such as Avibacterium paragallinarum Staphylococcus chromogenes, Ornithobacterium rhinotracheale, Mycoplasma synoviae, Mycoplasma gallisepticum, Gallibacterium anatis, and Escherichia coli among others), fungus and eukaryotic organism (Eimeria) that may affect lesions and survival. Information on key parameters that would be important to interpret the results, including the flock’s health status and the individual bird’s immune status preliminary to infection, is missing. Furthermore, the health status of flocks under field conditions is likely to be affected by nutrition, environmental stress, temperature, water etc. A lack of control on the environmental conditions, may have also had a masking or confusing effect on the heritability values.

Briefly, the authors infected a large (but perhaps not sufficient) number of birds that had a poorly characterized genetic makeup and had an unknown history of infections with an unknown inoculum using highly irreproducible method of infection to obtain heritability scores. It is not surprising that this study did not produce a reliable outcome in terms of heritability. A change in the goals and conclusions of the manuscript toward a simple descriptive characterization of the outcomes during NDV infection is suggested. This study may be the first to analyze a large number of birds of local African chicken ecotypes following “natural contact exposure” to velogenic NDV, therefore the findings on survival time, lesion scores of various organs, antibody levels in blood, viral load in tears and cloacal swabs may provide a useful baseline for other studies.

Author Response

Manuscript entitled “Genetic analyses of response of local and Ghanaian Tanzanian chicken ecotypes to a natural challenge with velogenic Newcastle disease virus” by Muhammed Walugembe and others analyze the outcomes of NDV infection on chickens from different ecotypes with the purpose of obtaining heritability estimates. The authors find that key traits are either not heritable or of low heritability, however they find some positive and negative correlations of the outcome with levels of infection, in agreement with previous knowledge of expected outcomes for NDV. The overall results (for example positive and negative correlations) are not revealing of any new pattern on NDV outcomes, and the low heritability observed may be explained by poor experimental design. Overall, it appears that too many experimental variables were allowed preventing the achievement of any meaningful conclusion. Surprisingly despite the weakness of the heritability evidence, the author concludes that “the response to mesogenic or velogenic NDV of these local chicken ecotypes can be improved by selective breeding”.

We understand the reviewers concern regarding the experimental design, however, the focus of the study was to mimic African rural farmers’ farms/backyards, where NDV is endemic and birds are exposed to different NDV strains over time, in order to maximize expression of genetic differences in resilience of unvaccinated local chickens to velogenic NDV. Some local farmers attempt to control NDV through vaccination, however, this approach has been deficient because of various reasons addressed in the current manuscripts and our previous published Lasota trial manuscripts. Because there are no previous holistic velogenic studies, it was important for us to understand the genetic basis of all feasible velogenic NDV response traits. As indicated in the background of this manuscript, experimental infection trials with velogenic strains are difficult and expensive because of the high biocontainment required. As a result, such experiments are often conducted with small sample sizes. We used acceptable sample sizes for genetic studies and used seeder birds to mimic exposure in the field to allow the determination of heritability for disease response traits across multiple replicates with potentially different NDV strains. Hence, our results are not specific to a single NDV strain, which is what would typically be used in controlled experimental infection trials. We believe the natural exposure trials that we used are the most feasible and relevant approach to conduct such experiments and study the various NDV response traits, with results that are directly applicable to the field. We acknowledge the low heritability estimates that were obtained for a number of the response traits that were investigated. However, survival time was actually quite heritable and that is the most relevant trait in terms of application. Hence, our overall conclusion that response to NDV can be improved by selective breeding was based on those results. We have revised the manuscript to further emphasize these points.

The heritability scores, including those obtained for lesions (0.11) were weak or moderate perhaps reflecting the experimental limitations of the study. Normally high heritability values of 0.5 mean that on average half of the differences among phenotypes of animals are genetic. Low values of approximately 0.1 mean that most of the differences are not genetic. For Tanzania, low to moderate estimates were found for natural antibody, survival time, proventriculus lesions scores, and pre-exposure growth rate. Estimates of heritability were in the same range for these traits for Ghana, except for natural antibody level, which was not heritable for Ghana. Heritability estimates for survival in the current study were greater than 0.10 for Ghana, Tanzania, and combined data analyses. Only proventriculus lesion score had heritability greater than 0.10 for Ghana, Tanzania, and combined data analyses among the lesion score traits. As the authors clearly state in the introduction, the determination of heritability for disease response traits is difficult due to the “potential uncontrolled” nature of infections among individual animals. A previous study by the same group described very low heritability of NDV response traits after infection with low pathogenic NDV strains in similar type of experiments [references 18] [19] [26]. In the study reported here, “natural exposure” trials were conducted with chickens from local ecotypes from Tanzania and Ghana as previously evaluated in [17], [18], and [28].  Surprisingly, no efforts were made here to eliminate or to reduce some known important factors that may lead to “potentially uncontrolled” outcomes. In fact, additional “uncontrolable” factors were introduced by the use of a challenge method “by contact” and by the use of a poorly characterized inoculum. The team appear to have ignored the fact that many genotypes of NDV viruses circulate in Africa simultaneously at one location, that these viruses have often high genetic and antigenic variability, that these viruses have low, medium and highly virulence, as no effort was made to purify or to genetically characterize the inoculum.

See response to the previous comment. Note also that heritability estimates of survival time were greater than 0.2 for each country and combined. This is considered sufficient to embark on genetic improvement, in particular with the use of genomics. Our previous studies reported moderate to high estimates of heritability of NDV response traits after infection with low pathogenic (vaccine/Lasota) NDV strains (references 17 and 18). The focus for this study was to understand the genetic basis of velogenic NDV response traits by mimicking the farmers’ backyard disease environment, where the spread of the virus is by “contact”. In addition, studying the genetics of response to NDV exposure with potentially different strains of the virus being present within and across replicates drives results to be relevant across strains, which is crucial for application in the field. Agreed, it would be useful to gain additional insight in the strains that were present in each trial. To that end, our group has also studied genotypes of NDV viruses that circulate in the study locations, and we have genetically characterized them (Reference 13). We agree with the reviewer that these viruses are medium or highly virulent. Further studies are ongoing to further understand the genetic basis of velogenic NDV exposures in local chicken ecotypes.  We have revised the manuscript (discussion section) to further address the point.

Several previous studies have indicated the existence of multiple bacterial, fungal and viral agents co-infecting poultry worldwide and this type of infectome is likely to be present in the inoculum and on the non-SPF birds used during the “natural” infectious process. The chicken infectome in Africa is not simply composed of avian influenza (the only other agent tested) but of a plethora of RNA and DNA viruses that are known to have immune suppressive effect (e.g., Gumboro), synergistic respiratory effects (e.g. Infectious Bronchitis), and opportunistic disease causing effects (e.g. bacteria in digestive, respiratory and immune tissues such as Avibacterium paragallinarum Staphylococcus chromogenes, Ornithobacterium rhinotracheale, Mycoplasma synoviae, Mycoplasma gallisepticum, Gallibacterium anatis, and Escherichia coli among others), fungus and eukaryotic organism (Eimeria) that may affect lesions and survival. Information on key parameters that would be important to interpret the results, including the flock’s health status and the individual bird’s immune status preliminary to infection, is missing. Furthermore, the health status of flocks under field conditions is likely to be affected by nutrition, environmental stress, temperature, water etc. A lack of control on the environmental conditions, may have also had a masking or confusing effect on the heritability values.

The birds used in the current experiment are progeny from the breeding flock that was raised in a controlled environment with ad-lib access to both feed and water. No clinical signs of disease were observed in the test birds, and freedom from avian influenza was tested in the seeder birds before their use to avoid introducing that other major respiratory virus into the study. The authors acknowledge the potential presence of other pathogens in the environment, and also that they may have varied between replicates. This variation is an important part of the experimental design, meant to estimate genetic parameters that are robust over a variety of ND viral strains, co-infecting pathogens and variation in environment; rather than narrowly defined conditions that, while a “cleaner” experimental design, do not yield information of broad application. We have addressed the reviewer’s concern in the discussion section.

Briefly, the authors infected a large (but perhaps not sufficient) number of birds that had a poorly characterized genetic makeup and had an unknown history of infections with an unknown inoculum using highly irreproducible method of infection to obtain heritability scores. It is not surprising that this study did not produce a reliable outcome in terms of heritability. A change in the goals and conclusions of the manuscript toward a simple descriptive characterization of the outcomes during NDV infection is suggested. This study may be the first to analyze a large number of birds of local African chicken ecotypes following “natural contact exposure” to velogenic NDV, therefore the findings on survival time, lesion scores of various organs, antibody levels in blood, viral load in tears and cloacal swabs may provide a useful baseline for other studies.

The birds in the current experiment are progeny from the breeder flock and initial characterization of the ecotypes was done in our previous vaccine/Lasota studies (References 17 and 18). We characterized the velogenic NDV genotypes (reference 7) and we believe an appropriate experimental design was used to study velogenic NDV response traits under natural exposure in the current study. The birds in the current experimental were genotyped, characterized and belong to defined ecotypes as addressed in the manuscript. This characterization agrees with the characterization of birds in our previous vaccine/Lasota trials (References 17 and 18). The authors do not consider the heritabilities to be unreliable. As expected, however, they are lower than could be found under more stringently controlled experimental conditions. In addition, similar heritabilities have been reported in an NDV-challenge experiment conducted in well-controlled conditions (ABSL-2 rooms, controlled temperature and lighting, consistent high-quality feed, one genetic line, and single stock of virus). Again, the goal for the current study was to generate this information for conditions that closely model the field conditions of small-holder farmers. Manuscript has been revised to address reviewer’s comment.

We agree with the reviewer that this study is the first to analyze a large number of African local chicken ecotypes following “natural contact NDV exposure”, and will be a very useful baseline for future studies. We have addressed reviewer’s comment in the conclusion.

Reviewer 3 Report

Newcastle disease in poultry is an important disease with a significant losses to the industry therefor genetic selection for some resistance to the disease is of high importance. The authors found various responses to the Newcastle disease virus (NDV), the disease causative agent, traits that were heritable in both Ghana and Tanzania chicken ecotypes. They provided evidence that virulent NDV response traits could be influenced through selective breeding. Their finding could be use in making breeding decisions in the selection of local ecotypes to be more resilient and survive when exposed to velogenic NDV 

Author Response

Thank you!

Round 2

Reviewer 2 Report

No comments.